# The Mechanical Investigation of Filament-Wound CFRP Structures Subjected to Different Cooling Rates in Terms of Compressive Loading and Residual Stresses—An Experimental Approach

**DOI:** 10.3390/ma14041041

**Published:** 2021-02-22

**Authors:** Wojciech Błażejewski, Michał Barcikowski, Marek Lubecki, Paweł Stabla, Paweł Bury, Michał Stosiak, Grzegorz Lesiuk

**Affiliations:** Faculty of Mechanical Engineering, Wroclaw University of Science and Technology, PL 50-370 Wroclaw, Poland; wojciech.blazejewski@pwr.edu.pl (W.B.); michal.barcikowski@pwr.edu.pl (M.B.); pawel.stabla@pwr.edu.pl (P.S.); pawel.bury@pwr.edu.pl (P.B.); michal.stosiak@pwr.edu.pl (M.S.); grzegorz.lesiuk@pwr.edu.pl (G.L.)

**Keywords:** composites, residual stresses, cooling rate, filament winding

## Abstract

Although cooling at ambient temperature is widely used and is said to be safe and convenient, faster cooling may have an influence not only on the time of the manufacturing process but also on the mechanical response, especially the residual stress. The study aimed to investigate the influence of the cooling rate after curing on the mechanical response of filament-wound thick-walled carbon fiber reinforced polymer (CFRP) rings. Three cooling rates were taking into consideration: cooling with the oven, at room temperature, and in the water at 20 °C. The splitting method was used to examine the residual strains. In the radial compression test, the mechanical response was investigated between the rings with different cooling regimes. The FEM analysis of the compression test in elastic range was also performed. Both the splitting method and the radial compression test showed no significant difference in the mechanical response of the CFRP rings. The presented results showed that the fast-cooling rate slightly decreases the mechanical performance of the filament-wound rings.

## 1. Introduction

The manufacturing of composite structures is inherently linked to the mechanism of residual stress (RS) forming. These stresses affect the mechanical properties of the manufactured composite structures, essentially, as evidenced in works [1,2,3,4,5,6]. In particular, this problem is strongly observed in the manufacturing of composite materials and their cylindrical structures commonly used in the oil and gas industry [7,8], the automotive sector including pressure vessels [9,10,11] as well as in the aerospace industry [12,13]. In recent decades a significant interest in the matter can be observed. In the study [14] it was found that the generation of residual stresses strongly fluctuates within the cooling rate. A higher cooling rate increases the initial residual stresses. Therefore, it was recommended to first reduce the cooling rate during the curing process to avoid possible cracks in the matrix when the ambient temperature is reached. In another work [15] the residual stresses were calculated for different cooling conditions at temperatures 0 °C and 25 °C. It has been shown that the significant differences in temperature between the curing and cooling conditions resulted in direct thermal shock, and a considerable amount of residual stress was created. It resulted in the formation of significant amounts of residual stress, which may accelerate the destruction of the structure. Samples cooled at 25 °C showed less residual stress than samples cooled at 0 °C.

The obtained results suggest the necessity of further work related to the assessment of the influence of carbon fiber reinforced polymer (CFRP) cooling conditions of structures on their mechanical properties. Based on these considerations, the key role is played by residual stresses, which can be estimated using experimental-numerical analysis.

In recent years, the subject of the study experienced a regression—especially since the available measurement and numerical technique [16,17] allows for efficient estimation of stress levels. One of the most widespread experimental methods is the slitting method [6,18,19,20] which is used as a primary investigation method in the presented paper. In the light of the works [14,15], the nature of the process optimization possibilities arises, not so much from a change in the isothermal conditions of the cooling, but from a process medium that allows receiving heat at different speeds, thus clearly influencing the overall cooling conditions. Recently, in the paper [20] a comparative analysis was performed, but only for flat components. However, the main aim of the study was to investigate the effect on fracture resistance expressed in the G_IC_ test. A positive impact of slow cooling has been demonstrated in increasing resistance against fracture.

Although there is a lack of such analyses in the literature for complex elements which are an elementary case of a cylindrical structure, this article attempts to bridge a literature gap in the field of the possible impact of cooling rate on cylindrical compressive strength of CFRP cylindrical structures. For this purpose, three types of cooling systems are investigated; quickly in the water at 20 °C (fast cooling, FC), in the air at 23 °C (intermediate cooling, IC), and slow cooling condition with an oven (slow cooling, SC).

## 2. Materials and Methods

Carbon fiber reinforced plastic composite cylinders were manufactured to investigate the mechanical response after different cooling regimes. The general information about the fiber and matrix is presented in Table 1 [21,22].

### 2.1. Temperature Gradient Evaluation

The first step was to check the temperature gradient across the cylinder wall cross-section during cooling. For this purpose, three cylinders with an inside diameter of 104 mm, a length of 400 mm, and a wall thickness of 19.2 mm were made, corresponding to 42 layers. During the winding, 6 thermocouples were placed at different wall depths. The arrangement of the sensors is shown in Figure 1.

Only the circumferential layers were wound up. The objects were made of Araldite LY 1564 epoxy resin with Aradur 3474 hardener reinforced with Torayca T720SC carbon fiber (material properties are presented in Table 2). All cylinders were cured in accordance with the manufacturer’s instructions for 5 h at a temperature of 80 °C. After this time, one of the cylinders was cooled quickly in water for 1 h at 20 °C (fast cooling, FC), the other was cooled for 5 h in air at 23 °C (intermediate cooling, IC), and the third was cooled as the oven cooled down for 12 h (slow cooling, SC). To simulate the conditions of cooling present in filament wound tanks (cooling from the outside layer in) specimens were cooled with a polyamide mandrel with a low thermal conductivity. The final product after the manufacturing process is shown in Figure 2.

### 2.2. Slitting Method—Experimental Analysis of Residual Strains

The slitting method was used to reveal the residual strains. The technique is easy to implement and involves cutting out a slice of the specimen to free the forces present in the composite after the manufacturing and cooling process [23,24,25]. In the method, the strains on the inner and outer surfaces of the specimen are measured. The specimen is mounted strictly and by use of a machining tool, such as a hacksaw, a piece of the ring is removed. In this investigation, strain gauges were mounted on the specimen in four locations. Two of them were on the inner circumference, and another two were on the outside circumference as can be seen in Figure 3.

To cut the ring a steel hacksaw was used. When the bending moment was large enough to close the ring, another cut next to the first one was performed to easily remove a slice of the specimen so that the slitting section was free. Photos of the ring before and after the experiment are shown in Figure 4.

### 2.3. Radial Compression Test

Five samples from fast and intermediate cooling and six from slow cooling were used in the radial compression tests. All samples were new and had no previous load history. The tests were carried out with the MTS 810 (load cell 100 kN) testing machine, at a compression rate of 3 mm/min. Figure 5 shows a diagram (a) of the test, as well as a view of the specimen mounted on a machine (b). The reaction force was measured.

### 2.4. Finite Element Method Analysis

The numerical analysis was performed in Abaqus/CAE environment using the linear elastic material model. The finite element method (FEM) analysis aimed to investigate whether the residual stresses may influence the material behavior in the elastic range. The material properties presented in Table 2 were obtained using the product datasheets [21,22] and the rule of mixture. They were set in a cylindrical coordinate system to reflect the hoop winding technology. Moreover, the calculated residual stresses were applied to the ring before the compression simulation was performed. The stresses were calculated using the linear dependence between measured strain signals and appeared stress level in the elastic range, namely, Equations (1) and (2) described in Section 3.4 were implemented.

The assembly in the simulation consisted of the ring and two plates: the lower one fixed and the upper with the vertical displacement. The plates were modeled as a rigid body as can be seen in Figure 6.

The ring was modeled as a 3D object consisting of 19,400 linear hexahedral elements of type C3D8R. The discrete model can be seen in Figure 7.

## 3. Results and Discussion

### 3.1. Temperature Gradient during Cooling

The diagrams of temperature in the cross-section of the cylinder wall during heating (Figure 8) and cooling in air (Figure 9) and in water (Figure 10) are presented below. Thermocouple 3 was damaged during the test and the data were discarded.

The maximum temperature difference between the outer and inner layers during heating was 20 °C, during cooling in air it was approximately 11 °C, and during cooling in water 42 °C.

The set temperature in the oven was obtained after 15 min whereas the temperature of the composite ring rose significantly slower and obtained the set 80 °C after about 3 h. Thermocouples 1 and 4 had the lowest temperature gain. The highest temperature gain was on the outer circumference of the composite element which indicates thermocouple 6.

In the case of air cooling, the highest temperature difference was about 11 °C. The highest temperature loss was obtained by thermocouple 6. The temperature in thermocouples 2 and 5, which were placed in the middle of the cross-section of the composite thickness, did not stand out significantly from the temperature on the inner surface of the cylinder.

For fast cooling in water, the highest temperature difference occurred, at the level of about 42 °C. A great temperature gradient was obtained. The outer surface of the cylinder cooled down significantly faster than the middle of the cross-section and the inner surface. Nevertheless, the difference between the temperature in the middle of the cross-section and the inner surface was also noticeable.

Unfortunately, the data recorded during the cooling in the furnace was corrupted. The time needed to reach room temperature, in this case, was about 12 h.

### 3.2. Radial Compression Test

As mention in the previous chapter, a radial compression test of the composite rings was performed. Figure 11, Figure 12 and Figure 13 show force-displacement diagrams obtained during radial compression of composite rings. All the figures show peaks and valleys characteristic of composite elements resulting from the destruction of individual layers of the material.

In the elastic part of the material behavior, there is great consistency between the samples. The mean value of the maximum force was 19.44 kN ± 1.08 kN. After the first falls, the composite ring could withstand a significant portion of energy since the curve did not fall dramatically. Generally, after the first drop, there is no rise in the force. However, in the case of specimen 6, the force rose even though small cracks occurred. This behavior is common in composite materials where single cracks may not affect the global strength of the component.

For the intermediately cooled (IC) rings there is a small, but noticeable difference in the elastic part of the experiment. The mean value of the maximum force was 20.11 kN ± 0.87 kN.

In the case of fast cooled (FC) rings the mean value of the maximum force was 17.84 kN ± 0.75 kN, which is significantly lower than in the previous two groups (8% in the case of slow cooling and 11% in the case of intermediate cooling).

Table 3 summarizes the maximum forces obtained for each of the samples as well as their mean values. Coefficient of variation was calculated to assess the dispersion of results. To compare the behavior of differently cooled rings in the elastic region, the radial stiffness was calculated. This procedure was performed based on regression analysis and calculation of the slope coefficient (in the range of 25–75% of the linear part). The average difference between FC and IC is about 5% and between FC and SC, it is 7%. However, taking into account the data deviation, there is no statistical difference between the specific cooling regimes in terms of the elastic region.

The damage behavior of the composite ring may be described mainly as delamination between the filament-wound layer. It can be observed in Figure 14. On the inner circumference, local buckling areas with fiber breaking also occurred. These processes generally took place after the maximum force.

### 3.3. FEM Analysis

In the FEM simulation, the force-displacement curve was obtained. Since no damage model was applied, only the elastic part can be discussed, where the agreement between the experiment and the simulation is satisfactory. The comparison is presented in Figure 15.

The curves for the three cooling regimes do not differ since residual stresses did not influence the elastic behavior of the composite rings under radial compression. The stress fields show the concentration of stress in the side areas of the ring, both tension and compression (Figure 16). These reflect the real damage mechanism, where in the inner circumference there were local buckling areas due to compressive loads.

The different cooling rates reflected in the residual stresses did not influence the stress distribution in the specimens. The only difference was the value of the stress at the set displacement of the plates. However, the difference did not exceed 5%.

### 3.4. Residual Strains Obtained by Slitting Experiment

To reveal the residual stresses after the manufacturing process the slitting method was used. Each ring was equipped with four strain gauges which were supposed to measure the strains in the hoop direction. Graphs showing the strains of rings during cutting are presented in Figure 17. It is noticeable that strains increased gradually, finally obtaining the maximum values. The final strains of the inner circumference (gauges 1 and 3) ended up being negative, whereas those on the outer circumference (gauges 2 and 4) had positive values.

Although the process of cutting was continuous, there are some plateaus areas on the graphs, as can be seen in Figure 17. It was by the fact that during the process of cutting the deformation had an abrupt character.

Judging from the residual strains, the residual stresses can be estimated using the following equations [16]:(1)$$\sigma_{\theta }^{in}=E_{\theta }\Delta \varepsilon_{C}^{in}$$
(2)$$\sigma_{\theta }^{out}=E_{\theta }\Delta \varepsilon_{C}^{out}$$
where $\Delta \varepsilon_{C}^{in}$ and $\Delta \varepsilon_{C}^{out}$ are the hoop strains on the inside and outside surfaces during the slitting method (radial-cut), respectively, and $E_{\theta }$ is the known elastic modulus. The results of the calculations are presented in Table 4.

## 4. Conclusions

The current study aimed to investigate the influence of the cooling rate on the mechanical response of the filament-wound composite cylinders. The following conclusions can be drawn from the performed research work and numerical analyses:

The radial compressive strength does not show significant differences between SC and IC cooling mode. In comparison with the obtained results, only FC exhibited the smallest value of maximum force recorded in the test for each specimen. The difference in the maximum force during radial compression test is 8% in terms of FC to SC comparison and 11% in terms of FC to IC comparison. This effect might be caused by microstructural changes (microcracks) induced during the fast-cooling regime. This hypothesis demands further investigation.

The lack of statistical difference between the different cooling regimes in terms of elastic stiffness during radial compression suggests that fast cooled thick-wall composite rings could perform equally in the elastic and safe range of the loading.

The analysis of the residual strain distribution and the residual stress values obtained suggest the selection of slow cooling as a method that ensures equally effective radial compressive strength as IC.

On the contrary, from the practical point of view, the fast-cooling regime does not decrease the mechanical durability of the composite cylinders in a way that disqualifies the method from further analyses. In industry, a slight drop of the strength may be acceptable in light of the possibility of a significant reduction in manufacturing time.

Although static loads seem to indicate quick cooling as the least advantageous treatment after the manufacturing process, it should be pointed out that this mechanism should be much more strongly emphasized in the studies related to the influence of both time and stress concentrators (fatigue and fracture). Furthermore, a comparison (despite similar static results) of IC and SC in fatigue tests seems to be fully reasonable in further work of the authors’ team.

## Figures and Tables

**Figure 1 materials-14-01041-f001:**
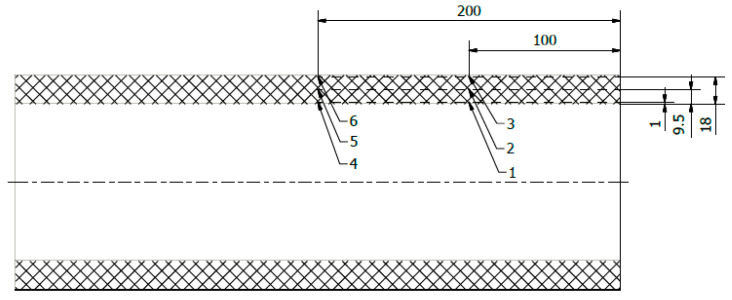
Diagram of the arrangement of temperature sensors (all dimensions in mm).

**Figure 2 materials-14-01041-f002:**
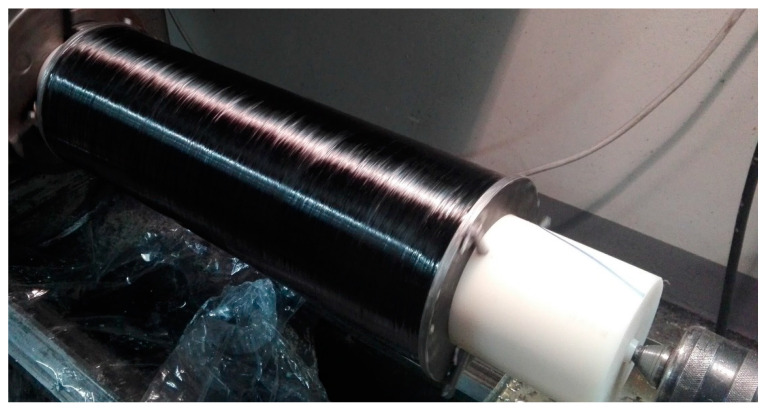
Filament wound cylinder subjected to the experimental campaign.

**Figure 3 materials-14-01041-f003:**
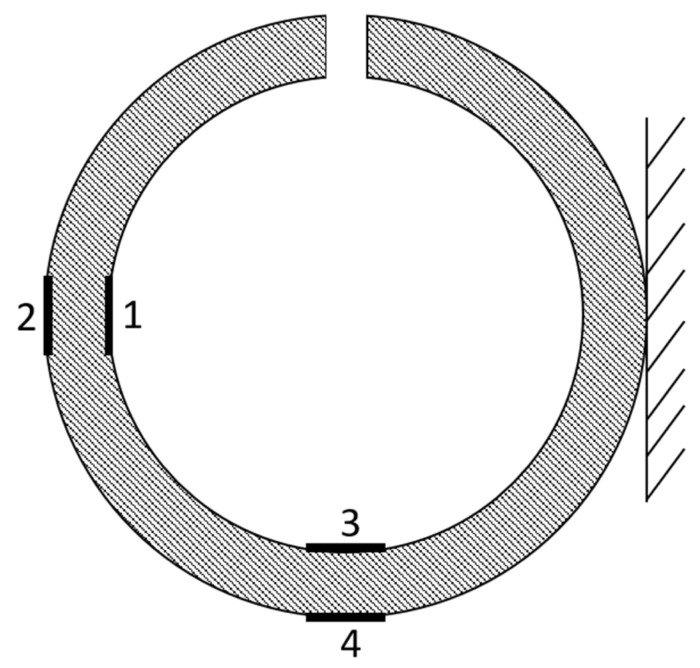
Diagram of the strain gauges’ locations in the slitting method.

**Figure 4 materials-14-01041-f004:**
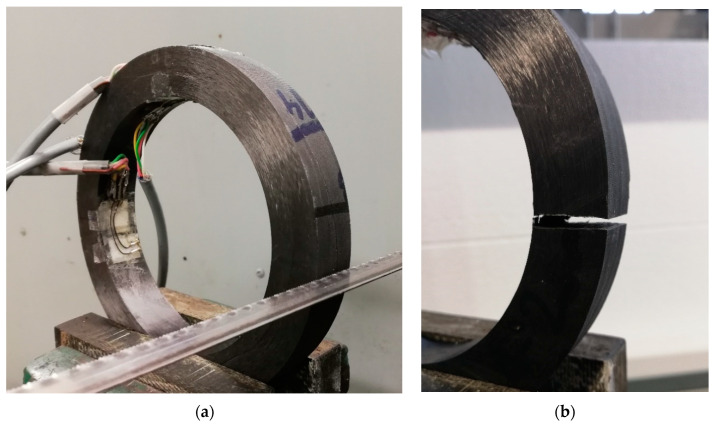
Photos of the tested ring: (**a**) before cutting and (**b**) after cutting.

**Figure 5 materials-14-01041-f005:**
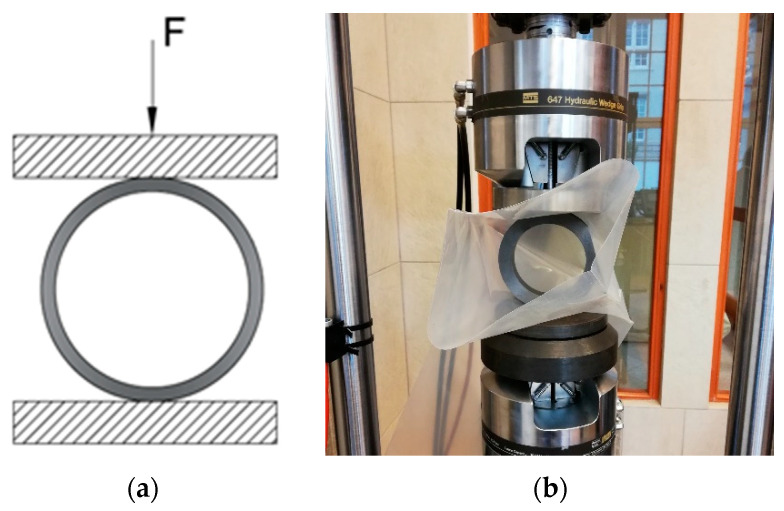
Radial compression of composite rings: (**a**) scheme of the experimental boundary conditions and (**b**) view of the sample mounted on the machine.

**Figure 6 materials-14-01041-f006:**
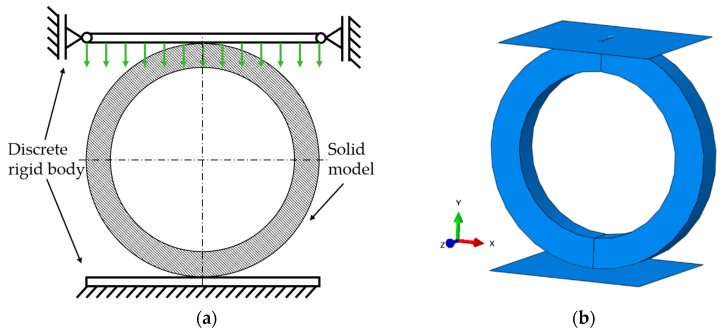
Boundary conditions (**a**) and geometric model (**b**) of the ring.

**Figure 7 materials-14-01041-f007:**
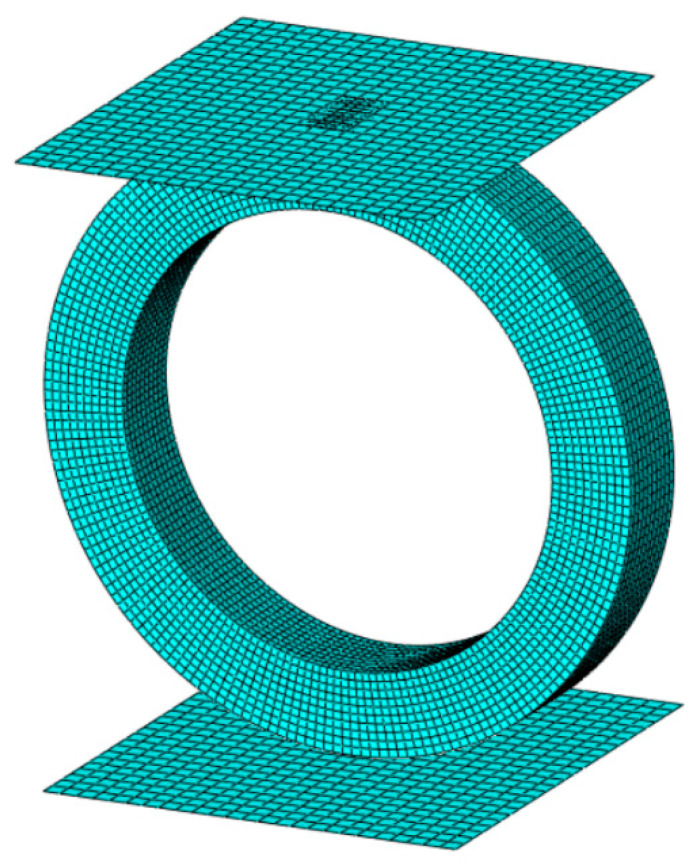
A discrete model of the analyzed ring.

**Figure 8 materials-14-01041-f008:**
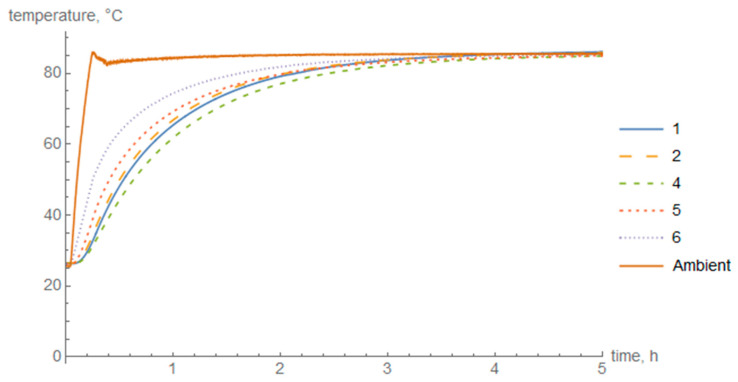
Temperature change in individual layers during cylinder heating.

**Figure 9 materials-14-01041-f009:**
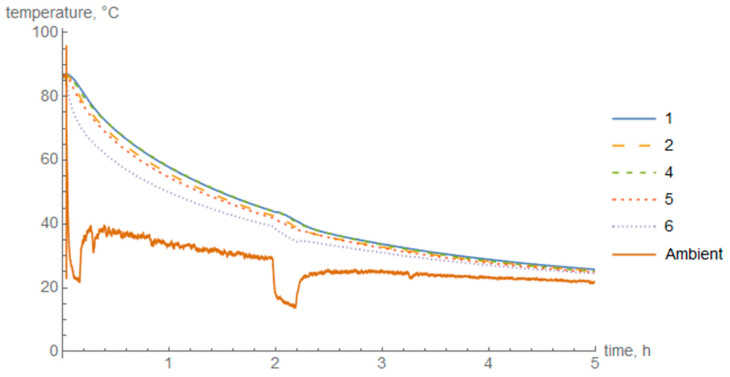
Temperature change in individual layers during air cooling.

**Figure 10 materials-14-01041-f010:**
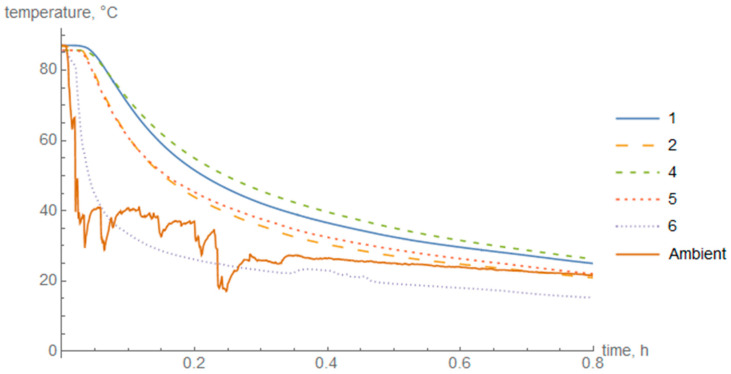
Temperature change in individual layers during cooling in water.

**Figure 11 materials-14-01041-f011:**
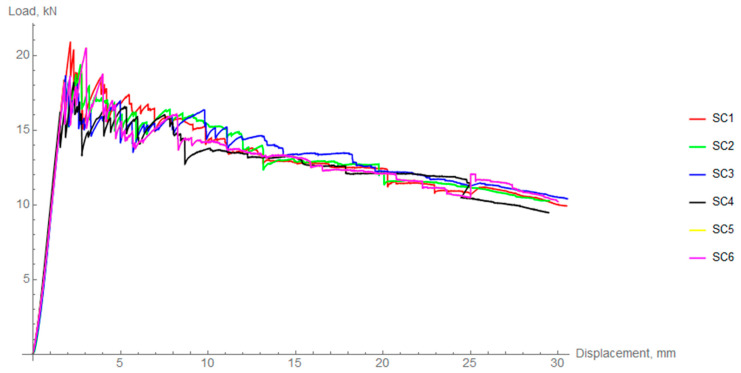
Load-displacement curves for radial compression of slow cooled (SC) rings.

**Figure 12 materials-14-01041-f012:**
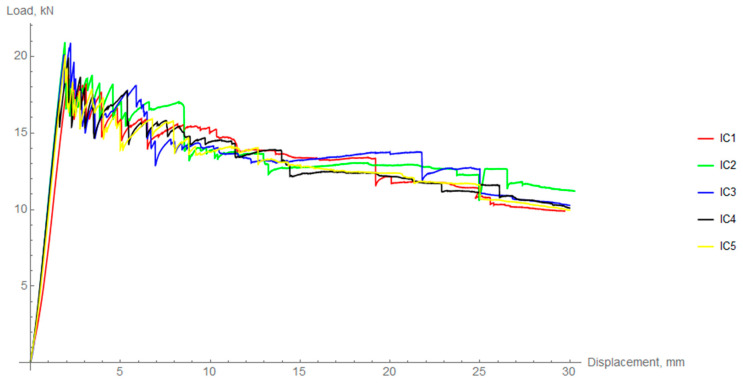
Load-displacement curves for radial compression of intermediately cooled (IC) rings.

**Figure 13 materials-14-01041-f013:**
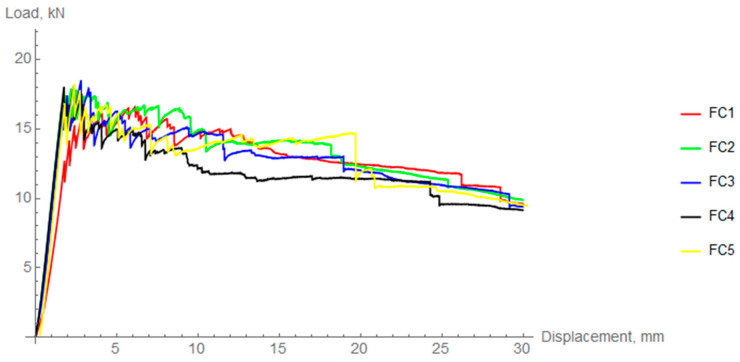
Load-displacement curves for fast cooled (FC) rings.

**Figure 14 materials-14-01041-f014:**
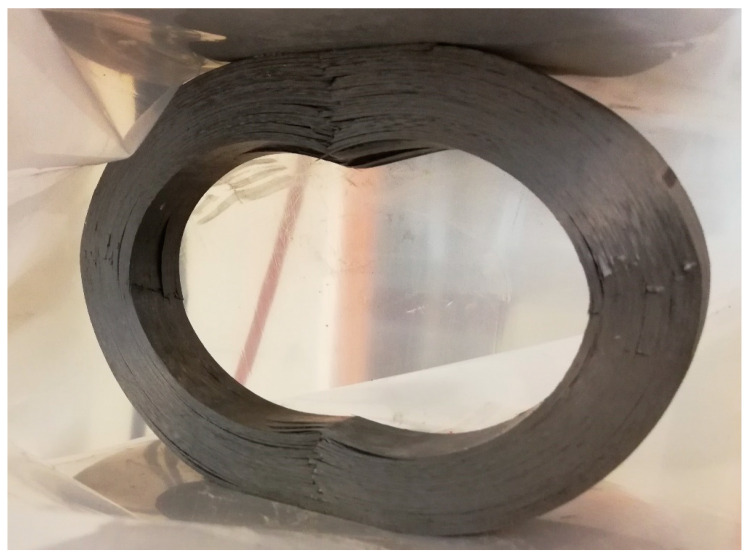
Composite ring after radial compression test.

**Figure 15 materials-14-01041-f015:**
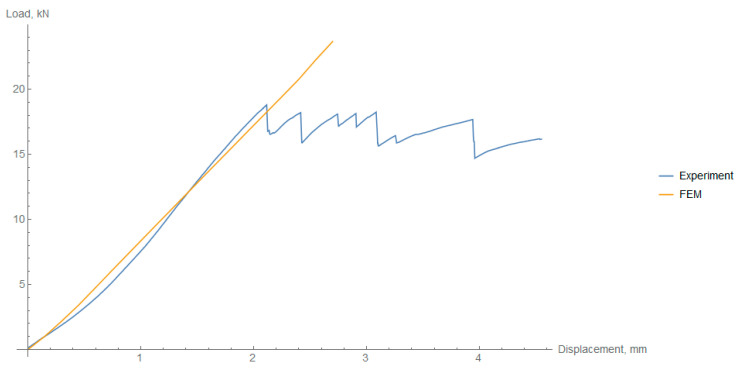
Comparison between experimental results of radial compression of rings and FEM analysis for the fast cooled rings.

**Figure 16 materials-14-01041-f016:**
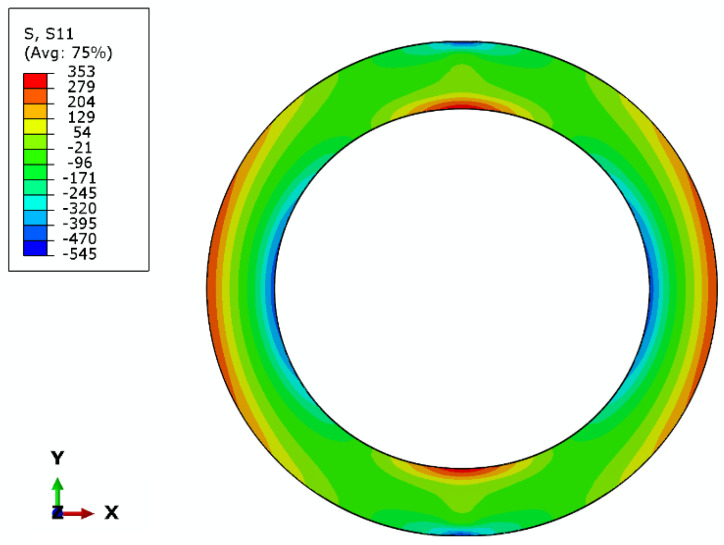
Stress fields in hoop direction (S11) for the fast cooled rings.

**Figure 17 materials-14-01041-f017:**
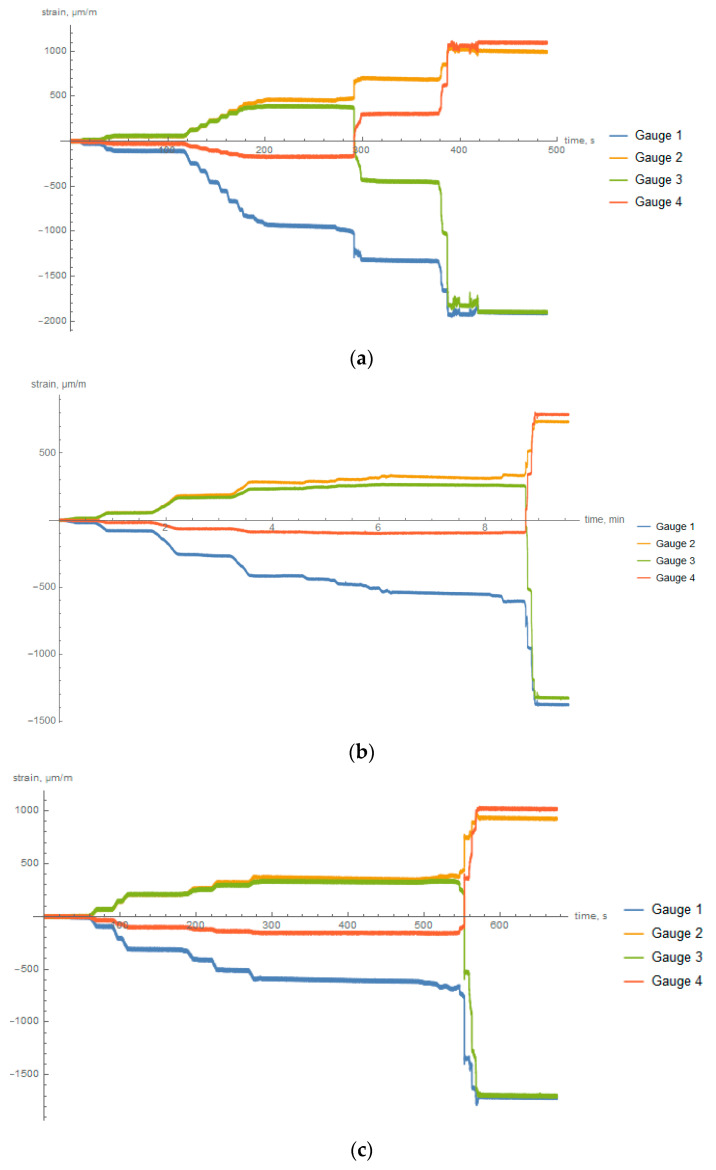
The strain of the composite rings during the slitting experiment, recorded on the inside and outside surfaces for: (**a**) slow cooled, (**b**) intermediately cooled, and (**c**) fast cooled rings.

**Table 1 materials-14-01041-t001:** Fiber and matrix characteristics.

Parameter	Value
Fiber characteristics	
Name	Torayca T720SC
Type	Carbon fiber
Filament	36k
Tensile strength	5880 MPa
Tensile modulus	265 GPa
Density	1.8 g/cm^3^
Matrix characteristics	
Resin name	Araldite LY 1564
Hardener name	Aradur 3474
Resin type	thermoset
Formulation	100:26
Pot life at 40 °C	60–70 min
Tensile strength	80 MPa
Tensile modulus	2900 MPa
Density	1.1 g/cm^3^

**Table 2 materials-14-01041-t002:** Material data used for FEM analysis.

E_1_ (MPa)	E_2_ (MPa)	E_3_ (MPa)	ν_12_	ν_13_	ν_23_	G_12_ (MPa)	G_13_ (MPa)	G_23_ (MPa)	ρ (g/cm^3^)
114,000	4988	4988	0.33	0.33	0.4	2152	2152	1076	1.5

**Table 3 materials-14-01041-t003:** Max loads for radial compression of rings.

Specimen	Max Compressive Load (kN)	Mean Compressive Load (kN)	CoV (%)	Radial Stiffness (N/mm)	CoV (%)
IC1	18.80	20.11 ± 0.87	4.32	11,377.5 ± 916.4	8.05
IC2	20.93				
IC3	20.88				
IC4	19.87				
IC5	20.08				
SC1	20.92	19.44 ± 1.07	5.50	11,560.5 ± 288.7	2.50
SC2	19.41				
SC3	18.68				
SC4	18.9				
SC5	20.52				
SC6	18.97				
FC1	16.55	17.84 ± 0.75	4.20	10,764.4 ± 1298.7	12.06
FC2	17.89				
FC3	18.49				
FC4	18.11				
FC5	18.18				

**Table 4 materials-14-01041-t004:** Estimated residual stresses in fiber direction on the inside and outside surfaces of the rings.

Cooling Regime	$\sigma_{\theta }^{in}\;(\mathrm{MPa})$	$\sigma_{\theta }^{out}\;(\mathrm{MPa})$
Intermediate cooling	−126	76
Slow cooling	−150	86
Fast cooling	−143	84

## Data Availability

Data sharing is not applicable to this article.

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
