# Peer review of "The Mechanical Investigation of Filament-Wound CFRP Structures Subjected to Different Cooling Rates in Terms of Compressive Loading and Residual Stresses—An Experimental Approach"

_materials, 2021, doi:10.3390/ma14041041_

Round 1

Reviewer 1 Report

Dear,

You have a lot of quantitative data, but you don't take into account in your discussions. Your experiments are good but the discussion are very ligth.

You have to argue more the discussion with quantitative datas and give the reason or hypotheses why the fast cooling have less properties than the IC and SC cooling.

Best regards

Author Response

Ms. Ref. No.: materials-1101210

Title: The mechanical investigation of filament-wound CFRP structures subjected to different cooling rates in terms of compressive loading and residual stresses – an experimental approach

Response to Reviewer #1

Dear Reviewer,

I would like to thank you for your meticulous review of our manuscript. Thank you for your time and effort.

1.
R:
 Line 41-42: Samples cooled at 25°C showed less residual stress than samples cooled at 25°C. I think this sentence is not correct maybe 0°C if the reference is [15]

A: Thank you for your perceptiveness, the error has been corrected in the text. The correct version is: “ Samples cooled at 25°C showed less residual stress than samples cooled at 0°C. “

2.
R:
 Line 63: and slow cooling condition with the(Slow Cooling, SC). The sentence is not finished. Oven?

A: Thank you for your excellent remark, the error has been corrected in the text. We added the word “oven” for clarification purposes

3.
R:
Can you said if the resin is a thermoset or a thermoplastic because the cooling process influence the crystallinity of resin for thermoplastics?

A: The resin used in the test is thermosetting. Information about this has been added in Table 1.

4.
R:
Line 82-84: Can you give the time of cooling to show the difference between fast cooling and slow cooling?

A: Water cooling lasted 1 hour, air cooling 5 hours, and cooling with the oven - 12 hours. Information on this subject has been supplemented in the text.

5.
R:
Can you explain how you cooled your cylinder, I think you have an aluminium or steel mandrel to realize the filament winding, did you remove it before cooling or not?

A: To simulate the conditions of cooling present in filament wound tanks (cooling from the outside layer in) specimens were cooled with a solid polyamide mandrel which thermal conductivity is low. Information on this subject has been supplemented in the text.

6.
R:
Explain that the compression test is realized on virgin ring and not on cutted ring.

A: Thank you for your perceptiveness. Yes, radial compression tests were performed on fresh rings without previous load history. This information has been added in the text.

7.
R:
Line 138: Not approx. but approximately.

A: Thank you for your remark. This has been corrected in the text.

8.
R:
Where is the diagram of cooling in the oven? Or if you don’t have, please indicate the time to reach the ambient temperature.

A: Unfortunately, the data recorded during the cooling in the furnace was corrupted. The time needed to reach room temperature, in this case, was about 12 hours.

9.
R:
In materials and methods, you said that the curing time is 5h at 80°C, but in the figure 8, we only show 3h from 75°C. What is the reason?

A: Thank you for the insightful question. In industrial applications, the curing time is measured as the time the element spends in the oven at given temperature without taking into account the time it takes the insides of the element to heat up to this temperature (as in most cases the internal temperature of the element cannot be known). Our experiment simulates, and thus reflects, the industrial practice. We have added this explanation in the materials and methods section.

10.
R:
As you explain in line 142 to 146, your curing time is only 2-3 hours. Do you think that your cylinder is correctly cured and that your matrix is reticulate?

A: Thank you for your fine question. Yes, to our knowledge, the curing time of the samples was sufficient to ensure complete cross-linking of the resin and the achievement of the desired mechanical properties. This would be the effect the manufacturer would achieve by following the resin curing guidelines.

11.
R:
In your figure 11, 12 and 13, the better information of your curves is at the beginning, elasticity and the first 5 mm of displacement. I think you have to plot the curve until 5 or 10 mm, to show the reproducibility of your 6 tests.

A: Dear Reviewer. Thank you for the suggestion. However, our intention was to show the overall compression behaviour of the material. We understand the reviewer's point, but the key for further work was to show the first maximum and stiffness and as such, this is shown in the tables below with statistical measures. We are very sorry if we deceived the reviewer, but we prefer to leave the whole graph - without multiplying the initial parts.

12.
R:
Line 168: In the elastic part of the material (between 0 and 15kN?)

A: Thank you for your excellent question. It was considered that the elastic range for the tested samples encompassed the range from the beginning of loading until the first cracks were registered. There is no plastic deformation in the material.

13.
R:
Line 169: The maximum obtained force oscillates around 20 kN. The mean value of maximum force is 19.4kN +/-1kN

A: Thank you for your remark. Information on the values of the maximum forces has been clarified in the text.

14.
R:
Please indicate the range measurement of the stiffness (between 0 and 0.5mm?)

A: In terms of the linear part of the slope of the elastic part before the nonlinearity occurred. This procedure was performed based on regression analysis and calculation of the slope coefficient ( in the range of 25%-75% of the linear part) - which is a measure of stiffness.

15.
R:
Line 172: Why the force rise for specimen 6, have you an idea?

A: This behaviour is common in composite materials where single cracks may not affect the global strength of the component.

16.
R:
Line 175-176: For the intermediately cooled (IC) rings there is a small, but noticeable difference in the elastic part of the experiment. The maximum force was around 20 kN. Please indicate quantitative data, you have the mean force and standard deviation and you have the mean stiffness and SD. You can also calculate the Coefficient of Variation as (SD/Mean)*100. For the stiffness you can show that the CoV is equal to 8% that is small in an experimental point of view. You can do the same for the 3 cooled experiment.

A: Dear Reviewer, Thank you for your suggestions. Of course - in the revised version of the manuscript - please note we have added these measures as well as relevant comments. Thank you again - we can see that after this addition the paper looks better analytically.

17.
R:
Line179: which is significantly lower than in the previous 2 groups.
Yes!!But give us the quantitative data the decrease is 10%...

A: Thank you for the suggestion. The differences in percentage were added to in the manuscript.

18.
R:
Table 3: You said that 6 specimens are tested for each configuration, why do you have only 5 in IC and FC? Said that at least 5 specimens are tested for each configuration, because for me one of the specimens has removed of the mean because the force is not enough....

A: First of all thank you for your perceptiveness.  We wrote imprecisely that in each case (SC, FC, and IC) we used 6 samples for the radial compression tests. After revision of the data, it turned out that for IC and FC these tests were performed only on 5 samples. The change was also introduced in the text.

19.
R:
Fig13: You have only 4 specimens and in the table 3 you have 5 value to calculate the mean.

A: One of the curves was missing due to an oversight when drawing the graph. The error has been fixed in the text.

20.
R:
Table 4: indicate the range of measurement of the stiffness.

A: Dear Reviewer, that is true – please note that an additional comment was added and also commented regarding REV comment #14 -Again, In terms of the linear part of the slope of the elastic part before the nonlinearity occurred. This procedure was performed based on regression analysis and calculation of the slope coefficient ( in the range of 25%-75% of the linear part) - which is a measure of stiffness.

21.
R:
You can group table 3 and 4.

A: All Authors agreed with Reviewer, now both tables are consolidated. Thank you again for such a suggestion.

22.
R:
To compare the behavior of the differently cooled rings in the elastic part, the radial stiffness was calculated and showed in Table 4. No significant difference was observed between the cylinders.

Between FC and SC, the difference is 10% in stiffness, your right it’s not significant but argue your point of view with quantitative data.

A: Dear reviewer, the average difference between FC and IC is about 5% and between FC and SC it is 7%. However, taking into account the data deviation, there is no statistical difference between the specific cooling regimes in terms of the elastic region.

23.
R:
In figure 16, you can add the real rupture of specimen show in fig.14 to show that your model is in accordance with the rupture of specimen.

A: Dear Reviewer. Thank you for this indication. The reviewer has reached out to a topic that is a distinct issue for a different paper. We are currently working on a damage model and description using decohesion for a much larger group and diversity of damage types. However, it is worth noting that our intention in this paper was only to address the elastic part and this fits with the main purpose of the paper. The presented damaged cylinders are of qualitative value. We think that the reviewer will soon be able to appreciate our other work devoted only to the issue of damage.

24.
R:
Figure 17: the legends are missing...

A: Thank you for the comment. We are sorry for any inconvenience you have experienced. The problem was due to incorrect graphics dimensions which has been corrected.

25.
R:
Why have we seen plateau? Maybe because you saw the ring in 4 times....

A: Plateaus in the graphs are caused by the fact that the change in the deformation of the rings occurred abruptly during their cutting. The cutting itself was carried out in a continuous process, once for each ring.

26.
R:
To argue the last section of slitting method, I think you can simulate a ring with a cutting part and impose a load to close the ring. With this simulation you can have the stress or stain imposed to close the ring and compare to the experiment.

A: Dear Reviewer, thank you for the suggestion. The idea of such a simulation is very interesting and worth considering. As mentioned above, we will work on more comprehensive FEM model in the future. In this work, we focused on the elastic part of the material behaviour in a radial compression test as it states in ASTM D2412.

27.
R:
You have a lot of quantitative data, but you don't take into account in your discussions. Your experiments are good but the discussion are very ligth.

You have to argue more the discussion with quantitative datas and give the reason or hypotheses why the fast cooling have less properties than the IC and SC cooling.

A: Thank you for the comment. The section has been extended.

Look forward to your favourable consideration

Reviewer 2 Report

This research paper investigates the effect of the cooling rate on the mechanical characteristics of the CFRPs.

The results are important for industrial use as it suggests an increase in the cooling step as there is a minor disadvantage of increasing the speed.

The experimental design and methodology are acceptable. English quality could be higher.

Below some corrections can be found:

Line 41: Samples cooled at 25°C showed less residual stress than samples cooled at 25°C. (Please correct)

Line 63: ...slow cooling condition with the oven.

Line 174: There is no IC6 displayed in Figure 12. (If failed, please explain in the text)

Line 178: There are no FC5 and FC6 samples displayed in Figure 13. (If failed please explain.) (Table 3 contains FC5) 

Table 188: Table 4... FC samples show very high standard deviation, maybe due to decreased number of samples. The authors might increase the numbers of the test to decrease it to a reasonable level.

Line 223: Non of the graphics has a legend next to it, please identify the lines.

Author Response

Ms. Ref. No.: materials-1101210

Title: The mechanical investigation of filament-wound CFRP structures subjected to different cooling rates in terms of compressive loading and residual stresses – an experimental approach

Response to Reviewer #2

Dear Reviewer,

I would like to thank you for your meticulous review of our manuscript. Thank you for your time and effort.

1.
R:
 The experimental design and methodology are acceptable. English quality could be higher

A: All Authors would like to thank you for this suggestion, the English language was double-checked by native-speaking person

2.
R:
 Samples cooled at 25°C showed less residual stress than samples cooled at 25°C. (Please correct)

A: Thank you for your perceptiveness, the error has been corrected in the text. The correct version is: “ Samples cooled at 25°C showed less residual stress than samples cooled at 0°C. “

3.
R:
 Line 63: ...slow cooling condition with the oven..

A: Thank you for your excellent remark, the error has been corrected in the text. We added the word “oven” for clarification purposes

4.
R:
 Line 174: There is no IC6 displayed in Figure 12. (If failed, please explain in the text)

A: First of all thank you for your perceptiveness.  We wrote imprecisely that in each case (SC, FC, and IC) we used 6 samples for the radial compression tests. After revision of the data, it turned out that for IC and FC these tests were performed only on 5 samples. The change was also introduced in the text.

5.
R
: Table 188: Table 4... FC samples show very high standard deviation, maybe due to decreased number of samples. The authors might increase the numbers of the test to decrease it to a reasonable level

A: Thank you for your excellent remark. Elements made of composite materials are inherently less homogeneous than those made of, for example, metallic materials. This could be one of the reasons for the elevated values of standard deviations. The highest standard deviation was noted for samples cooled in water (FC). Due to the rapid course of the cooling process, there are suspicions that this could lead to internal cracks which led to a further reduction in the homogeneity of the material. Please take note that the number of FC samples was not smaller than the number of IC samples. We believe it is probable that larger number of samples would not improve the standard deviation, especially so because the further samples would have to be taken from other parts of the manufactured cylinder, and therefore might have experienced differing thermal history.

6.
R
: Line 223: None of the graphics has a legend next to it, please identify the lines.

A: Thank you for the comment. We are sorry for any inconvenience you have experienced. The problem was due to incorrect graphics dimensions which have been corrected.

Look forward to your favourable consideration

Round 2

Reviewer 1 Report

Dear Authors,

you improve your article.

best regards